# Development of Er^3+^, Yb^3+^ Co-Doped Y_2_O_3_ NPs According to Yb^3+^ Concentration by LP–PLA Method: Potential Further Biosensor

**DOI:** 10.3390/bios11050150

**Published:** 2021-05-11

**Authors:** Cheol-Woo Park, Dong-Jun Park

**Affiliations:** 1College of Engineering, San Diego State University, 5500 Campanile Drive, San Diego, CA 92182, USA; cpark4@sdsu.edu; 2Division of Advanced Materials Science and Engineering, Hanyang University, Seoul 04763, Korea; 3Department of Surgery, University of California San Diego 212, Dickinson Street, MC 8236 CTF B R310, San Diego, CA 92103, USA; 4Department of Otorhinolaryngology, Yonsei University Wonju College of Medicine, 20 Ilsan-ro, Wonju 26426, Korea

**Keywords:** diagnostic material, Y_2_O_3_:Er^3+^, Yb^3+^, nano ceramic particles, up-conversion luminescence, liquid phase–pulsed laser ablation

## Abstract

As diagnostic biosensors for analyzing fluids from the human body, the development of inorganic NPs is of increasing concern. For one, nanoceramic phosphors have been studied to meet the increasing requirements for biological, imaging, and diagnostic applications. In this study, Y_2_O_3_ NPs co-doped with trivalent rare earths (erbium and ytterbium) were obtained using a liquid phase–pulsed laser ablation (LP–PLA) method after getting high density Er, Yb:Y_2_O_3_ ceramic targets by Spark plasma sintering (SPS). Most NPs are under 50 nm in diameter and show high crystallinity of cubic Y_2_O_3_ structure, containing (222), (440), and (332) planes via HR–TEM. Excitation under a 980 nm laser to a nanoparticle solution showed 525 and 565 nm green, and 660 nm red emissions. The green emission intensity increased and decreased with increasing Yb^3+^ additive concentration, when the red spectrum continuously strengthened. Utilizing this study’s outcome, we suggest developing technology to mark invisible biomolecules dissolved in a solvent using UC luminescence of Er^3+^, Yb^3+^ co-doped Y_2_O_3_ NPs by LP–PLA. The LP–PLA method has a potential ability for the fabrication of UC NPs for biosensors with uniform size distribution by laser parameters.

## 1. Introduction

For years, emerging technologies for detecting hazardous substances, using NPs, have been developed [1,2]. It was reported that inorganic-based NPs for in vitro diagnostic technology induce physical bonding, depending on the components of the solvent and the substrate [1]. In particular, in the field of biosensors, chemical and structural bonding is being proposed, albeit in a new direction, and developing differentiated materials is required [2,3]. Numerous researchers are attempting to combine the fields of nanoparticle sizes, properties, drug detection, and hazardous substance selections; the development of diagnostic sensors using ceramic-based NPs is one research field [4,5]. The up-conversion (UC) phenomenon generally refers to a process in which photons are converted from a low-energy state to a high-energy state via multistep absorptions, such as excitation state absorption (ESA) and energy transfer UC (ETU) from infrared or near-infrared (NIR) light. A typical activator of UC is the Er ion, which has various energy levels due to 4f electrons, and shows green and red emissions [6]. This represents an IR absorption at approximately 980 nm in ^5^F_5/2_ → ^5^F_7/2_ transfer and plays an important role in increasing the UC intensity via energy transfer [7]. In particular, UC phosphors doped with Yb^3+^ and Er^3+^ ions are promising materials for multiplex labeling due to their ability to emit light in various colors [8], which can be controlled by adjusting the amount of doped ions and particle sizes under a single 980 nm IR source. UC matrices, which use Yb^3+^ and Er^3+^ ions, have mostly been studied as fluorides, such as NaYF_4_ [9], YF_3_ [10], and LiYF_4_ [11]. Although they are the most efficient matrix materials due to their strong UC luminescence intensity and good photostability, some researchers are concerned about the fluoride toxicity of UC phosphors. Alternatively, yttrium oxide (Y_2_O_3_) has been extensively researched as it has high chemical and thermal stability and is a typical UC matrix material with a relatively low phonon energy [12].

These UC phosphors have applications as laser materials [13], nano bio-labels [14], nano-lighting devices [15], nano 3D display devices [16], bio-imaging [17], and photodynamic treatment [18], because their particles are nanosized. Bio-imaging and cancer treatment using UC NPs is being studied, especially regarding their low absorption in near-infrared areas, which are called optical windows. Absorption of living cells at near-IR light is low, while that at visible and far-IR light is high owing to hemoglobin and water, respectively. Hemoglobin and water are predominant substances found in large amounts throughout the human body [6,19]. Therefore, the use of UC nanoparticles can utilize examination of the abdominal cavity through bio-images without requiring a biopsy. The emission of UC phosphors is proportional to the number of photons required to release an exciting source power: L ∝ P_n_ (L = emission inertia, n = number of photons required to emit, and P = exciting source power) [18]. Therefore, it is essential to investigate the power dependence and particle size effects of UC phosphors to obtain important insights for the study of common phosphors and other doped materials.

Among UC nanophosphor manufacturing methods, sol–gel synthesis [20] and hydrothermal synthesis [21] methods have been primarily studied. However, these methods have several shortcomings, such as low yields and surface contamination from solvents [17,18]. Because of during the chemical preparation process, the contamination caused by chemical precursors or additives may lead to quenching, surface fouling, and toxicity stem from residuals. The use of LP–PLA is another UC nanophosphor manufacturing method. It is based on a phenomenon in which chemical species, such as atoms, molecules, ions, radicals, clusters, and electrons, are emitted from the surface of a target material irradiated by a laser beam focused with an energy density above a specific threshold. This method is generally applied to metal-based materials; however, some researchers have applied it to optical substances, such as fluorescent substances and inorganic compounds [22]. It can easily produce nanoparticle-dispersed solutions with high bioaffinity and multielemental composite NPs [23,24]. In addition, it allows highly efficient collection of NPs because all produced NPs are captured by the solvent.

In this study, Er^3+^, Yb^3+^ co-doped Y_2_O_3_ UC NPs of 50 nm or less were successfully produced using LP–PLA, and the microstructure and optical properties of the specimen were analyzed. In particular, the change in emission intensity at wavelengths (545 and 655 nm) excited at 980 nm by Yb^3+^ ions added in varying concentrations, the output intensity relative to the incident beam intensity, and the corresponding emission mechanism was investigated in detail. In addition, we suggest the development of technology using UC luminescence of Er^3+^, Yb^3+^ co-doped Y_2_O_3_ NPs by LP–PLA to mark dissolved biomolecules dissolved in a solvent. This LP–PLA method potentially has the ability for fabrication of UC NPs, for a biosensor with uniform size distribution, by laser parameters (power, fluence, wavelength, pulse duration), types of targets, and liquid media, such as urine, plasma, etc. Thus, the LP–PLA is believed to be a promising approach to produce various novel UC phosphor nanomaterials in order to detect UC luminescence, used for the biosensor, capable of detecting biomaterials and harmful substances dissolved in various solvents, and performing fluorescent indicators.

## 2. Materials and Methods

### 2.1. Er, Yb:Y_2_O_3_ Target Preparation

High purity Y_2_O_3_ (99.99% pure, Cenotec, Haman, Korea), La_2_O_3_ (99.9% pure, Kojundo, Japan), Er_2_O_3_ (99.9% pure, Rare Metallic Co., Ltd., Chiyoda-Ku, Japan), and Yb_2_O_3_ powders (99.9% pure, Shin-Etsu Chemical, Tokyo, Japan) for commercial applications were used as starting materials. The Yb_2_O_3_ powder was mixed with 1 mol% of Er_2_O_3_, and 0–5 mol% of Yb_2_O_3_. This powder was then ball-milled with zirconia balls with a diameter of 1.5 mm at a weight ratio of 1:10 for 24 h. Thereafter, it was dried for 24 h in an electric oven at 60 °C. All the dried powders were put into a graphite die with an internal diameter of 15 mm that was wrapped with graphite foil; the die was then sintered using a spark plasma sintering (SPS) system (S-515S, Sumitomo Coral Mining, Ehime, Japan). The sintering conditions were as follows: (i) the entire process was conducted at 10 Pa vacuum and pressure of 30 MPa; (ii) the die was heated at 100 °C/min from room temperature to the final sintering temperature (1625 °C) for 15 min; (iii) after sintering, the pressure was released at 1100 °C and the current was cut off. In the sintered specimen, the surfaces of both sides were ground to diameters of 15 mm and thickness of 1 mm using a grinder, and polished with a 1 μm diamond paste for 1 h. The crystalline phases of those targets were identified by X-ray powder diffraction (XRD), operating at 40 kV using Cu-Ka radiation (λ = 1.5406 Å).

### 2.2. Fabrication of Er, Yb:Y_2_O_3_ NPs

The completed Er, Yb:Y_2_O_3_ target (relative density >99%) was added to a glass container filled with ethanol (99.5%, Daejung Chemicals & Metals, Siheung-si, Korea) and was ablated using the YAG:Nd laser (30 Hz repetition rate, pulse width 5–7 ns, maximum output of 60 mJ/pulse). The laser was placed at 250 mm from the target to form a dot with a diameter of 1 mm on the surface of the Er, Yb:Y_2_O_3_ specimen. During the ablation, the specimen was constantly rotated to prevent deep ablation. Laser ablation of the Er, Yb:Y_2_O_3_ target was performed for 1 h at room temperature. Figure 1f illustrates the laser ablation process in liquid medium for preparation of Er, Yb:Y_2_O_3_ nanocrystals.

### 2.3. Characteristic Analysis of Er, Yb:Y_2_O_3_ NPs

The resulting nanoparticle solution was placed on a carbon-coated copper grid, and a transmission electron microscope (TEM) (JEM 2100F, JEOL, Tokyo, Japan) was used to observe the microstructure, particle size, and shape. The size of the nanocrystals was statistically analyzed using 200 NPs in an TEM image. The luminescent spectra were recorded using a spectrophotometer at room temperature using the nanoparticle solution. A 980 nm laser diode with maximum 100 mW output was used for the excitation source (MDE981H8, Latech Co., Seoul, Korea). A photoluminescence spectrometer (PL) (LS 55, Perkin-Elmer, Waltham, MA, USA) was used to observe fluorescence properties in the range of 300–800 nm under 980 nm laser excitation.

## 3. Results

### 3.1. Crystal Structure Analysis and Preparation LP–PLA Targets

The intrinsic properties of NPs, possibly controlling particle size and combined with other materials by the LP–PLA method have been reported in far-reaching applications, ranging from material science to vectors for drug and gene delivery [25]. Laser synthesis of colloids, powered by robust, high-power lasers, has appeared to be a key process in enabling chemical stability and eco-friendliness for industrial manufacturing of functional nanomaterials. In light of those facts, we made Er, Yb:Y_2_O_3_ targets with varying concentrations of Yb^3+^ using the SPS to make NPs with diameters smaller than 50 nm, which can be synthesized with urine and body fluids by LP–PLA.

The XRD patterns obtained for the Er, Yb:Y_2_O_3_ targets, sintered with different Yb^3+^ ion concentrations (0, 1, 3, and 5 mol%) are shown in Figure 2a. All specimens matched the Er_2_O_3_ phase, Yb_2_O_3_ phase, or standard cubic Y_2_O_3_ (JCPDS Card No 41-1105) phase without other impurities, and Er^3+^ 1 mol% and Yb^3+^ (0–3 mol%) perfectly matched space group Ia-3. As shown in Figure 2b, when the peak of the (222) plane was magnified and examined, it was found that the peak shifted to a higher angle as the doping concentration of Yb^3+^ increased. This indicates that because the ion radius (R = 0.85 Å) of Yb^3+^ was smaller than the ion radius (R = 0.89 Å) of Y^3+^ in the Y_2_O_3_ ceramic matrix, the increasing amount of Yb^3+^ caused the reduction of unit cells. Those fabricated targets provide proof of having a high density with transparency after the polishing process (Figure 2c). Those Er, Yb:Y_2_O_3_ targets with various Yb^3+^ concentrations were observed to exhibit fluorescence when viewed by the naked eye. They emitted strong UC fluorescence, each with different luminescences, according to Yb^3+^ concentration under the 980 nm laser (Figure 2d). It means that we have successfully fabricated ceramic targets with a stoichiometric composition ratio.

### 3.2. Morphology of NPs by LP–PLA

Er, Yb:Y_2_O_3_ NPs according to varying concentrations of Yb^3+^, obtained by irradiating (Nd:YAG laser: 50 mJ/pulse) the laser on the target were investigated with regard to their particle shape and size distribution by TEM (Figure 3b,d). The form and size of all the produced NPs appeared similar, regardless of Yb^3+^ concentration, and were clearly spherical. As shown in the particle size distribution chart at the bottom right, particle sizes mostly ranged from 5 to 10 nm. Several medium-sized NPs (<100 nm) existed, but large sub-micrometric particles (<1 μm) were not found. Minjung Cho et al. showed that a fluorescent dye, of various nanosized silica particle suspensions, was intravenously injected into mice to identify toxicity and tissue distribution, in vivo. Incidence and severity of inflammatory response was transiently increased with injection of 200 and 100 nm silica NPs. However, there was no response related to injection of 50 nm particles [26]. Consequently, for biomedical applications, particles smaller than 50 nm are required. Therefore, we analyzed two groups (<50 nm and >50 nm) depending on particle size. The crystal morphology, crystal size, and crystallinity were observed more closely by the high-resolution TEM (HR-TEM).

Preferentially, we analyzed the mostly existential large number of spherical NPs of 5–10 nm (Figure 4a,b). They displayed cubic crystals, and, because of their high ionic properties, they were slightly faceted even at a very small size. The crystal planes were (222), (232), and (440) planes, and the grid spacings were about 0.306 nm, 0.224 nm, and 0.178 nm, respectively, which belong to a cubic crystal system. The Fast Fourier Transform (FFT) pattern in the highlighted area was an orderly array of hexagons and squares corresponding to a highly crystalline property, and represented a spot pattern corresponding to cubic Y_2_O_3_ [27,28]. On the other hand, the produced Er, Yb:Y_2_O_3_ NPs with a size larger than 50 nm showed some interplanar distances are about 309 nm, which correspond to the (222) plane, the same crystal plane with different angles was found in many areas at the same time. In addition, the FFT images showed a spot pattern in which the previously observed array of hexagons and squares overlapped. Given that these same crystal surfaces are found within a particle, it is believed that several small particles produced during laser irradiation have fused and condensed among themselves; thus, forming medium-sized NPs (<100 nm) [29].

According to previous research of LP–PLA [30,31], the initial stage of the interaction between the pulsed laser beam and the target surface generates a hot-plasma plume over the laser spot on the ceramic target (Figure 3a). The plasma is confined in the liquid during the laser ablation, it expands outside adiabatically at a supersonic velocity creating shock waves in front, which induces elevated pressure and further increase of plasma temperature. The interaction of laser beam with the surface of the target leads to vaporization of the surface and generates an ablation plume, which contains many species, such as atoms, clusters, and ions, rushing out with high kinetic energy.

The interaction of a laser beam with the surface of the target leads to vaporization of the surface and generates an ablation plume, which contains many species, such as atoms, clusters, and ions, rushing out with high kinetic energy. After the collapse of the plasma plume, the particles can interact with radiation. This can lead to generation of the large size particles by small nanoparticles as a result of their melting and defragmenting (Figure 4c) [32,33]. Moreover, this process can form new compounds containing atoms from both the target and the liquid by these intermediate species in the plume collide [34], and react with molecules of the surrounding liquid through electrons attached, which become negatively charged and attract ions in the plasma [35]. Therefore, the major advantage of this method is that it permits the preparation of coated colloids of uniform layers of diverse composition by the self-assembly method, which is based on electrostatic attraction.

### 3.3. Luminescence on Er, Yb:Y_2_O_3_ NPs

Those NPs were analyzed for UC luminescence for visible light areas of Er, Yb:Y_2_O_3_ with different Yb^3+^ concentrations under a 980 nm laser in the range of 500–700 nm (Figure 5a). Laser irradiation on the specimen produced strong green emissions of 525 and 565 nm and a red emission of 660 nm (Figure 5b). The 525 and 565 nm green emissions correspond to ^2^H_11/2_, ^4^S_3/2_ → ^4^I_15/2_ energy transfer of Er^3+^ ion, and the 660 nm red emission corresponds to ^4^F_9/2_ → ^4^I_15/2_ energy transfer [36] (Figure 5c). This corresponds to an UC process in which Er^3+^ absorbs 980 nm of low energy and emits 525, 565, and 660 nm of high energy. Doping Yb^3+^ showed a significant increase in emission intensity in all areas. The green emission was the strongest in the specimen with added 3 mol% Yb^3+^, but the red emission intensity increased more strongly in proportion to the Yb^3+^ concentration. This suggests that a high concentration of Yb^3+^ causes cross relaxation between ions and reduces the green emission intensity [37]. The inset diagram shows the CIE 1931 diagram for the color change of UC luminescence by Y_2_O_3_:Er^3+^ and Yb^3+^ NPs excited at 980 nm as the Yb^3+^ doping concentration increased. As the red element in the luminescence color became stronger according to the Yb^3+^ concentration, the luminescence changed from vivid green at 0 mol% to orange at 5 mol%.

The principle of UC emission when 980 nm energy was applied to the Er^3+^ and Yb^3+^ ions is shown in Figure 5d. Er^3+^ ions absorb photons with an energy of 980 nm, resulting in the GSA corresponding to ^4^I_15/2_ → ^4^I_11/2_. Under 980 nm excitation, ESA corresponding to ^4^I_11/2_ → ^4^F_7/2_ by another absorption of 980 nm or multi-phonon relaxation corresponding to ^4^I_11/2_ → ^4^I_13/2_, followed by ESA corresponding to ^4^I_13/2_ → ^4^F_9/2_ could be possible. The electrons excited to ^4^F_7/2_ level generate non-radiant relaxation, falling to ^2^H_11/2_, ^4^S_3/2_, and ^4^F_9/2_ level, which causes the emission of energy and produces electromagnetic waves corresponding to 522, 548, and 656 nm, matching with the relaxation to the ground state [38]. In the case of doping Yb^3+^ ions, the GSA corresponding to ^2^F_7/2_ → ^2^F_5/2_ could be possible under 980 nm excitation. At this point, the energy of Yb^3+^ can be transferred to Er^3+^, by relaxing electrons in ^2^F_5/2_ to ^2^F_7/2_ level. This transferred energy is approximately matched with the excitation of ^4^I_11/2_ → ^4^F_7/2_ or ^4^I_13/2_ → ^4^F_9/2_, enhancing the emission of Er^3+^ [39]. However, excess concentration of Yb^3+^ would interrupt the excitation of electrons in Er^3+^, that is, cross relaxation between ^4^F_7/2_ → ^4^I_11/2_ of Er^3+^ and ^2^F_7/2_ → ^2^F_5/2_ of Yb^3+^ could also possible. After then, electrons in ^4^I_11/2_ level receives energy from electrons in the ^4^F_7/2_ level of other Er^3+^, resulting in ^4^I_11/2_ → ^4^F_9/2_ and ^4^F_7/2_ → ^4^F_9/2_, respectively. This process finally enhances the 656 nm emission and decreases the intensity of 522 and 548 nm emission [7].

## 4. Discussion

The yttrium oxide (Y_2_O_3_) NPs developed in this study have high dielectric constant and thermal stability, making it easy to develop NPs using lasers [1]. It is widely introduced as a potential material that can be applied to cancer treatment and biosensors using various biological imaging, photodynamic therapy, and phosphors [5]. The Y_2_O_3_ NPs produced by us generate fluorescence according to the Yb^3+^ concentration [40], and when a solvent is placed in body fluid or urine and a laser is generated, the positively charged particles of the solvent bind to the NPs [41] (Figure 6a). We propose that through purification of NPs, trace amounts of harmful substances can be measured by absorbance values (Figure 6a(i)). In addition, NPs generated from ceramic plates have negative charges, and ionized heavy metals [42] can be obtained usefully from sludge [2] (Figure 6a(ii)).

Bolotsky et al. [43] reported a study of the potential of healthcare and diagnostic medicine using ceramic NPs through electromagnetic analysis. Therefore, in regard to the Y_2_O_3_ NPs, we identified the light stability and the potential ability of the particles to bind to the ionized material, and are expected to be applied in the biosensor field, as they can be subjected to fluorescence analysis.

Our study attempted to develop fluorescent NPs (Y_2_O_3_ NPs), which could potentially be used to detect invisible biomolecules dissolved in a solvent (Figure 6a). The advantage of Y2O3 NP was developed to quickly recognize dissolved biomaterials. (Urine, Plasma etc.), prior to chemical analysis, such as spectrophotometer or chromatography (Figure 6c). The surface of the synthesized NPs is expected to be surrounded by harmful substances dissolved in a solvent. Y_2_O_3_ NPs can be synthesized and separated in stages with a small amount of biomaterial dissolved in a solvent (Figure 6c), so we expect to be able to compare the concentration of the biomaterial according to the difference absorbance of the NPs. Generally, Y_2_O_3_ NPs can be synthesized with glycoproteins or lipids dissolved in biological body fluids, and the harmful substances can be detected in fluorescence by indicating the concentration of toxic substances dissolved in urine (Figure 6a,b). In addition, a small number of harmful substances can be detected by synthesizing and separating toxic substances, such as heavy metals and tetracyclines dissolved in wastewater into NPs (Figure 6a(ii)). Therefore, our fluorescent Y_2_O_3_ NPs have the advantage of predicting diseases by detecting biomaterials and harmful substances dissolved in various biological solvents, and this technology is expected to be applied to a wide range of fields as a biosensor.

## 5. Conclusions

In this study, UC NPs were produced via the LP–PLA method using Er^3+^, Yb^3+^ co-doped Y_2_O_3_ of a high density (relative density >99%) as targets. The produced UC NPs, regardless of the concentration of Yb^3+^ added, were mostly 5 to 10 nm in size and were shaped like spheres with crystallization of a cubic Y_2_O_3_ structure. The up-conversion luminescence properties for 565 and 660 nm, investigated using the 980 nm laser diode in the photoluminescence spectrum, showed stronger luminescence from the substance as the Yb^3+^ concentration increased. When its concentration was above 5 mol%, the green emission of Er^3+^ drastically reduced and the red emission became significantly stronger. Furthermore, we suggest a biosensor technology that allows precise analysis and measurement of harmful substances present in body fluids or wastewater by Er^3+^, Yb^3+^ co-doped Y_2_O_3_ NPs. We propose that there is potential for application in medical fields and environmental sewage treatment fields by measuring different fluorescence according to Yb concentration.

## Figures and Tables

**Figure 1 biosensors-11-00150-f001:**
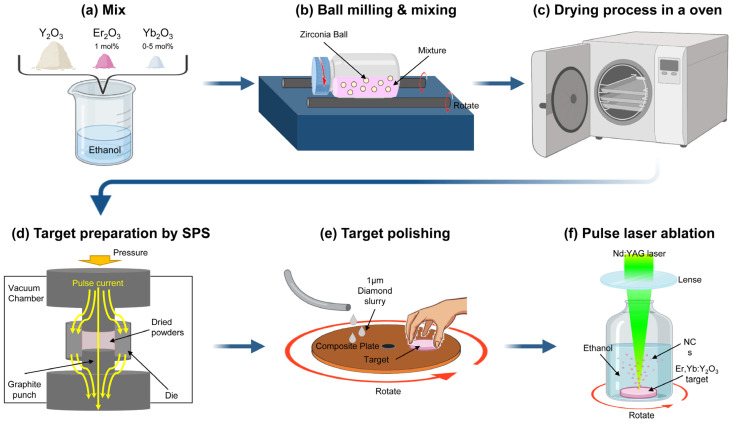
Illustration of process-manufacturing Er, Yb:Y_2_O_3_ NP methods.

**Figure 2 biosensors-11-00150-f002:**
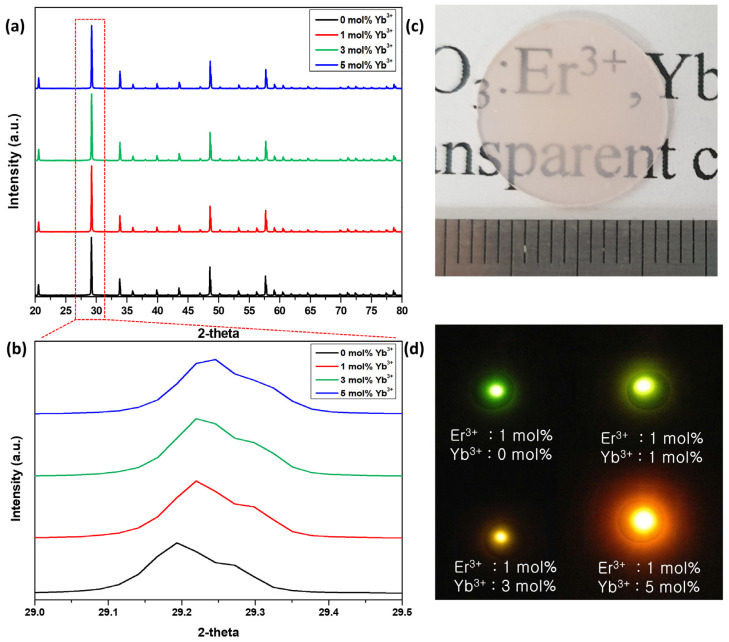
X-ray diffraction patterns of bulk Er, Yb:Y_2_O_3_ target containing 0 mol%, 1 mol%, 3 mol%, and 5 mol Yb^3+^ in the range of (**a**) 20–80° and (**b**) 29.0–29.5°; (**c**) photograph of Er, Yb:Y_2_O_3_ targets; (**d**) luminescence by 980 nm laser irradiation.

**Figure 3 biosensors-11-00150-f003:**
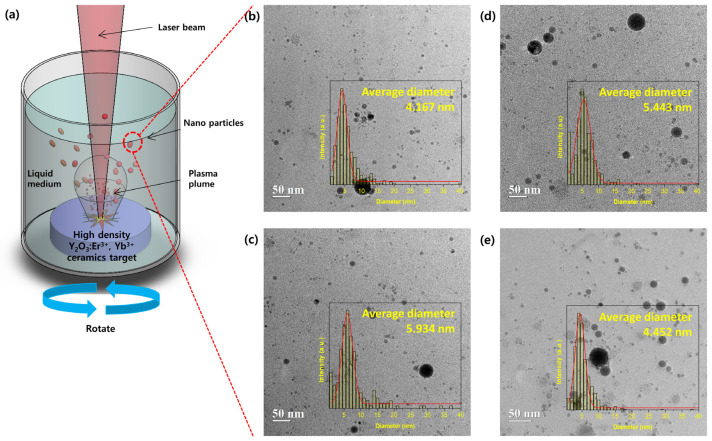
(**a**) Illustration of the laser ablation process, manufacturing NPs in liquid medium, TEM image of ablated Er, Yb:Y_2_O_3_ NPs with Yb^3+^ concentration 0 mol% (**b**), 1 mol% (**c**), 3 mol% (**d**), 5 mol% (**e**). The inset diagram shows the size distribution of the current image.

**Figure 4 biosensors-11-00150-f004:**
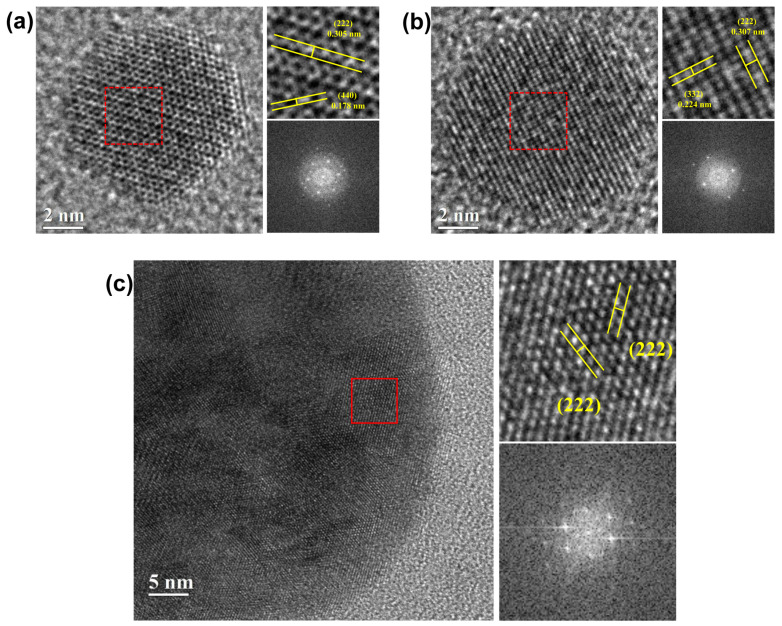
The high resolution TEM image of Er, Yb:Y_2_O_3_ NPs and its fast Fourier transformation image, (**a**) showing (222) plane with 0.307 nm and (332) plane with 0.305 nm and (440) plane with 0.178 nm d-spacing, and (**b**) (222) plane with 0.224 nm d-spacing, (**c**) 60 nm radius, showing (222) plane with 0.306 nm and 0.309 nm d-spacing.

**Figure 5 biosensors-11-00150-f005:**
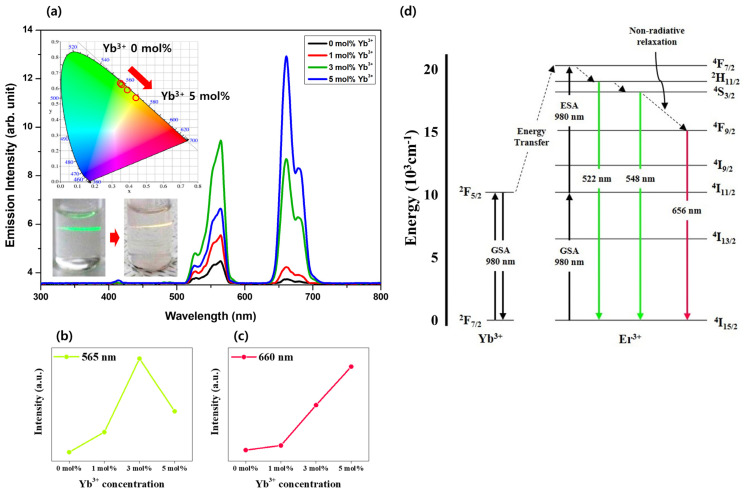
Plot of photoluminescence emission spectra of Er, Yb:Y_2_O_3_ NPs excited under 980 nm laser in the range of 500–700 nm (**a**). The 565 nm (**b**) and 660 nm (**c**) emission intensity vs. Yb^3+^ doping concentration plot was placed for each figure. (The inset diagram shows CIE 1931 diagram UC emission from Y_2_O_3_:Er^3+^, Yb^3+^ NPs excited under 980 nm, varying Yb^3+^ doping concentration.) (**d**) Schematic diagrams of the energy level of Yb^3+^, Er^3+^ in Y_2_O_3_ host matrix and the UC luminescence mechanism under 980 nm excitation and the process of 656 nm emission enhancement with energy-transferring among Er^3+^ ions in Yb^3+^-rich environment.

**Figure 6 biosensors-11-00150-f006:**
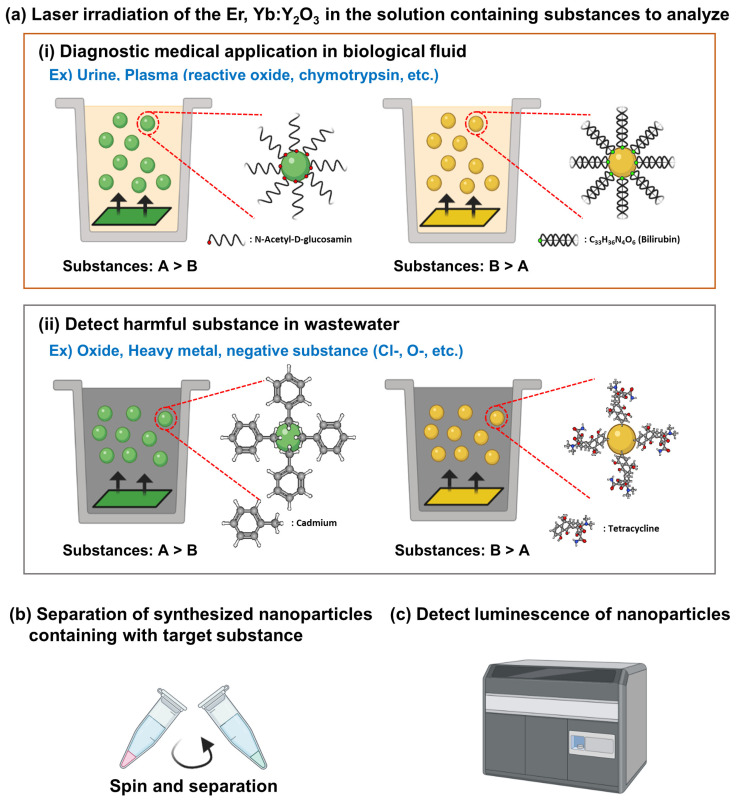
Summarized the application field of diagnosis medical and detecting harmful substances by different concentration Yb^3+^ of NPs with molecular interaction by surface charged properties.

## Data Availability

The data presented in this study are available upon request from the corresponding author.

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
