# Peer review of "Development of Er^3+^, Yb^3+^ Co-Doped Y_2_O_3_ NPs According to Yb^3+^ Concentration by LP–PLA Method: Potential Further Biosensor"

_biosensors, 2021, doi:10.3390/bios11050150_

Round 1

Reviewer 1 Report

The manuscript entitled “Development of Er3+, Yb3+ Co-Doped Y2O3 NPs according to Yb3+ Concentration by LP-PLA Method: Potential Further Biosensor” describes the fabrication of yttrium oxide (Y2O3) nanoparticles that are co-doped with Er3+ and Yb3+ via liquid-phase pulsed laser ablation (LP-PLA) and there applications as biosensors utilizing the upconversion (UC) phenomenon. The authors found that excitation of these nanoparticles at 980 nm led to green emissions at 520 and 565 nm, as well as a red emission at 660 nm. As the Yb3+ concentration was increased, the green emission intensity increased and subsequently decreased (beyond a Yb3+ concentration of 5 mol%), while the red emission increased continuously. The authors suggest that the LP-LPA method could provide an effective approach for the fabrication of UC nanoparticles with uniform diameters that would be suitable for biosensing applications.

    The research described in this manuscript is interesting and should have relevance in a variety of fields, such as inorganic chemistry, nanoscience, materials science, analytical chemistry, as well as the biomedical field. The nanoparticles synthesized by the authors were well-characterized by the authors via a variety of techniques, such as TEM, X-ray diffraction, and photoluminescence spectroscopy. The fabrication of the NPs via the LP-LPA method described by the author is also of interest.

    While the manuscript is generally clear, it could benefit from a significant amount of polishing, as there seem to be a number of issues with regard to grammar and sentence flow.

Aside from polishing of the writing, which will be required, I have only some minor suggestions. For example, I believe that it would be helpful to provide error margins for numerical values, to help provide a measure of the reliability and statistical significance of the numerical values that are presented, such as the grid spacings mentioned in lines 178, 179, 180, 182, 185, 186, and 190.

As mentioned, the results presented by the authors were generally well-characterized. Possibly it could also be beneficial to obtain elemental analysis measurements of the nanoparticles reported in this study.
In the discussion section (Section 4), it is mentioned that the yttrium oxide nanoparticles could potentially be used for biomedical and analytical applications such as recognition of compounds in biological fluids as well as the detection of harmful substances in wastewater. While I think that this would be a valuable use of this technology and that it would be feasible, it seems to have been proposed rather than executed in this work. With this in mind, it may be helpful if the authors provided a demonstration of this proposed use of these nanoparticles, such as performing an analysis of sample compounds in biological samples and waste water to provide a proof of concept. However, this may not be strictly necessary in this work (I would consider it an optional suggestion rather than a required revision), although it could be helpful. If it is not possible to perform these experiments, it could be something to consider as a future work for a future submission.

    Overall, I believe that this manuscript is suitable for publication pending minor revisions, and some suggestions in this regard are provided below.

Line 7, (address): Possibly “Dickinson street” should be changed to “Dickinson Street”.

Line 15: The phrase “been increased to meet the sensing” is unclear.

Line 16: Provide a definition for the abbreviation “RE’s”.

Lines 40-41: the phrase “via photon multi-steps such as” is unclear.

Lines 50-51: The phrase “as they are less than 980 nm to visible light” is unclear.

Lines 82-83: The sentence “As well as, we are trying to suggest that the development of technology to mark invisible biomolecules dissolved in a solvent using UC luminescence of Er3+, Yb3+ co-doped Y2O3 NPs by LP-PLA” is unclear.

Lines 119-120: The sentence “The laser ablation process in liquid medium for the preparation of Er, Yb: Y2O3 nanocrystals.” is unclear.

Line 129: “Photoluminescence Spectrometer” can be changed to “A photoluminescence spectrometer”.

Line 139: “by LP-PLA method” can be changed to “by the LP-PLA method”.

Line 142: “eco-friendly” can possibly be changed to “eco-friendliness” or “providing eco-friendliness”.

Line 144: “to make under the 50 nm size NPs which” can be changed to “to make NPs with diameters smaller than 50 nm which” or “to obtain NPs with diameters smaller than 50 nm which”.

Line 147: The phrase “and a second phases” is unclear.

Line 156: “observed to fluorescence by the naked eye” can possibly be changed to “observed to exhibit fluorescence when viewed by the naked eye” or  “observed to be fluorescent when viewed by the naked eye”.

Line 156-157: the phrase “They have strong lights” is unclear.

Line 168: “were investigated particle shape and” can possibly be changed to “were investigated with regard to their particle shape and”.

Line 182: The phrase “There are also showed 60 nm radius, showing (222) plane with” is unclear.

Lines 182, 183, 185, 186, and 190: Possibly error margins should be provided for the numerical values of the grid spacing mentioned here. 

Line 198: “generates hot-plasma plume” can be changed to “generates a hot-plasma plume”.

Line 210: The phrase “by that the self-assembly method” is unclear.

Line 245: “floor state” can be changed to “ground state”.

Lines 277-278: “which to detect invisible biomolecules dissolved in a solvent” can be changed to “which could potentially be used to detect invisible biomolecules dissolved in a solvent”.

Lines 278-279: The phrase “The advantage of Y2O3 NPs was developed to recognize quickly bio-substance dissolved” is unclear.

Line 303: “a technology of biosensor” can possibly be changed to “a biosensor technology”. 

Line 304: “substances presents in” can be changed to “substances present in”.

Author Response

Q1. Line 7, (address): Possibly “Dickinson street” should be changed to “Dickinson Street”.

Response: Thank you for your comment. We have changed the “Dickinson street” to “Dickinson Street”

Q2. Line 15: The phrase “been increased to meet the sensing” is unclear.

Response: We appreciate your careful reading. We have changed the sentences containing the phrase “been increased to meet the sensing”.

“Among them, nanoceramic phosphors have been studied to meet the increasing requirement for biological, imaging and diagnostic applications.” : Line 15.

Q3. Line 16: Provide a definition for the abbreviation “RE’s”.

Response: Thank you for your comment. We have changed the term “RE’s” to “rare earths”.

Q4. Lines 40-41: the phrase “via photon multi-steps such as” is unclear.

Response: We appreciate your careful reading. To clear, we have revised the phrase “via photon multi-steps such as” to “via multistep absorptions such as”.

Q5. Lines 50-51: The phrase “as they are less than 980 nm to visible light” is unclear.

Response: Thank you for your careful reading. We tried to describe that the fluorides are excellent host materials in the field of UC phosphors. We have changed the sentences containing the phrase “as they are less than 980 nm to visible light”.

“Although they are the most efficient matrix materials due to their strong UC luminescence intensity and good photostability, some researchers are concerned about the fluoride toxicity of UC phosphors.” : lines 50-51.

Q6. Lines 82-83: The sentence “As well as, we are trying to suggest that the development of technology to mark invisible biomolecules dissolved in a solvent using UC luminescence of Er3+, Yb3+ co-doped Y2O3 NPs by LP-PLA” is unclear.

Response: Thank you for your comment. We have changed the sentence.

“As well as, we are trying to suggest that the development of technology using UC luminescence of Er3+, Yb3+ co-doped Y2O3 NPs by LP-PLA to mark invisible biomolecules dissolved in a solvent." : Lines 82-84.

Q7. Lines 119-120: The sentence “The laser ablation process in liquid medium for the preparation of Er, Yb: Y2O3 nanocrystals.” is unclear.

Response: Thank you for your comment. We tried to show how to make the Er, Yb: Y2O3 nanocrystals by laser ablation process in liquid medium. We have changed the sentences.

“Figure 1f illustrates the laser ablation process in liquid medium for preparation of Er, Yb: Y2O3 nanocrystals.” : Lines 119-120.

Q8. Line 129: “Photoluminescence Spectrometer” can be changed to “A photoluminescence spectrometer”.

Response: Thank you for your comment. We have changed the “Photoluminescence Spectrometer” to “A photoluminescence spectrometer”

Q9. Line 139: “by LP-PLA method” can be changed to “by the LP-PLA method”.

Response: Thank you for your comment. We have changed the “by LP-PLA method” to “by the LP-PLA method”.

Q10. Line 142: “eco-friendly” can possibly be changed to “eco-friendliness” or “providing eco-friendliness”.

Response: Thank you for your comment. We have changed the “eco-friendly” to “eco-friendliness”.

Q11. Line 144: “to make under the 50 nm size NPs which” can be changed to “to make NPs with diameters smaller than 50 nm which” or “to obtain NPs with diameters smaller than 50 nm which”.

Response: Thank you for your comment. We have changed the “to make under the 50 nm size NPs which” to “to make NPs with diameters smaller than 50 nm which”.

Q12. Line 147: The phrase “and a second phases” is unclear.

Response: We appreciate your careful reading. We have changed the sentences containing the phrase “and a second phases”.

“The XRD patterns obtained for the Er, Yb:Y2O3 targets, sintered with different Yb3+ ion concentrations (0, 1, 3 and 5 mol%) are shown in Figure 2a.” : Line 145-147.

Q13. Line 156: “observed to fluorescence by the naked eye” can possibly be changed to “observed to exhibit fluorescence when viewed by the naked eye” or “observed to be fluorescent when viewed by the naked eye”.

Response: Thank you for your comment. We have changed the “observed to fluorescence by the naked eye” to “observed to exhibit fluorescence when viewed by the naked eye”.

Q14. Line 156-157: the phrase “They have strong lights” is unclear.

Response: Thank you for your comment. We have changed the sentences containing the phrase “They have strong lights”.

“They emitted strong UC fluorescence each with different luminescences according to Yb3+ concentration under the 980 nm laser (Figure 2d).” : Line 157-158.

Q15. Line 168: “were investigated particle shape and” can possibly be changed to “were investigated with regard to their particle shape and”.

Response: Thank you for your comment. We have changed the “were investigated particle shape and” to “were investigated with regard to their particle shape and”.

Q16.Line 182: The phrase “There are also showed 60 nm radius, showing (222) plane with” is unclear.

Response: We appreciate your careful reading. We removed the sentences containing the phrase “There are also showed 60 nm radius, showing (222) plane with”.

Q17. Lines 182, 183, 185, 186, and 190: Possibly error margins should be provided for the numerical values of the grid spacing mentioned here. 

Response: Thank you for your comment. We have changed these sentences as this follow:

“They displayed cubic crystals, and, because of their high ionic properties, they were slightly faceted even at a very small size. The crystal planes were (222), (232), and (440) planes, and the grid spacings were about 0.306 nm, 0.224 nm, and 0.178 nm, respectively, which belong to a cubic crystal system. The Fast Fourier Transform (FFT) pattern in the highlighted area was an orderly array of hexagons and squares corresponding to a highly crystalline property and represented a spot pattern corresponding to cubic Y2O3 [22,23]. On the other hand, the produced Er,Yb:Y2O3 NPs with a size larger than 50 nm showed some interplanar distances are about 309 nm, which correspond to the (222) plane, the same crystal plane with different angles was found in many areas at the same time.” : Lines 182, 183, 185, 186, and 190.

Q18. Line 198: “generates hot-plasma plume” can be changed to “generates a hot-plasma plume”.

Response: Thank you for your comment. We have changed the “generates hot-plasma plume” to “generates a hot-plasma plume”.

Q19. Line 210: The phrase “by that the self-assembly method” is unclear.

Response: We appreciate your careful reading. We have changed the sentences containing the phrase “by that the self-assembly method”.

“Therefore, the major advantage of this method is that it permits the preparation of coated colloids of uniform layers of diverse composition by the self-assembly method, which is based on electrostatic attraction.” : Line 207-210.

Q20. Line 245: “floor state” can be changed to “ground state”.

Response: Thank you for your comment. We have changed the “floor state” to “ground state”.

Q21. Lines 277-278: “which to detect invisible biomolecules dissolved in a solvent” can be changed to “which could potentially be used to detect invisible biomolecules dissolved in a solvent”.

Response: Thank you for your comment. We have changed the “which to detect invisible biomolecules dissolved in a solvent” to “which could potentially be used to detect invisible biomolecules dissolved in a solvent”.

Q22. Lines 278-279: The phrase “The advantage of Y2O3 NPs was developed to recognize quickly bio-substance dissolved” is unclear.

Response: Thank you for your comment. We have changed the “The advantage of Y2O3 NPs was developed to recognize quickly bio-substance dissolved” to “The advantage of Y2O3 NP was developed to quickly recognize dissolved biomaterials.” : Line 361-362

Q23 Line 303: “a technology of biosensor” can possibly be changed to “a biosensor technology”. 

Response: Thank you for your comment. We have changed the “a technology of biosensor” to “a biosensor technology”.

Q24 Line 304: “substances presents in” can be changed to “substances present in”.

Response: Thank you for your comment. We have changed the “substances presents in” to “substances present in”.

Reviewer 2 Report

Abstract:

Define RE's (I assume "rare earths") and UC (I assume "up-conversion")

Introduction:

Line 33: what does "physical bonding" mean? Is this physisorption as opposed to chemisorption?

Line 51: What is meant by the concerns with fluoride toxicity "as they are less than 980 nm to visible light"? Do the rare-earth fluorides need to be excited by visible light that releases toxic fluoride ions?

Line 59: with regard to "optical windows", it would be helpful to add why the NIR range is useful in terms of tissue absorption and the potential for in vivo measurements.

Line 67: What are typical surface contaminants and do they cause toxicity or other detrimental effects? It would be good to highlight problems with surface contamination to further motivate the use of PLA for synthesis.

Line 69: What does "gravitational atoms" mean?

Line 74-76: I recommend citing some literature in which PLA nanoparticles are used for biological/medical applications to show that the use of PLA nanoparticles is well established.

Line 83: Some examples of "invisible biomolecules" should be provided. As written, it's unclear exactly what UC NPs could detect that other methods can't.

Materials and Methods

Line 115: "reserve rate" should be changed to "repetition rate"

Line 116: What is the unit "md" for laser intensity? Usually that should be in W cm^-2 or as fluence in J cm^-2

Results

Line 139: Should add references to the statement about what PLA nanoparticles are used for.

Line 149: Looking at Figure 2b, it appears that the most up-shifted peak is observed with 1% Yb doping, whereas 3% and 5% are at lower 2theta. Thus the statement that the peak upshifts with increasing Yb content is not consistent with the presented data. Please explain.

Figure 3: mean particle size with standard deviation should be added to panels b through e

Line 173: No NPs >50 nm are included in the size distributions in Figure 3, although some of the TEM images do show them. It would be helpful to clarify that these NPs are rare and motivate why their detailed structure is worth investigating.

Figure 4: What was the Yb doping of the NPs shown in the images?

Line 191: Some explanation should be added for why a range of diffraction angles for the (222) plane is observed. Does this have anything to do with the hypothesis that the large NPs come from small NPs being sintered together during PLA?

Line 245: "floor state" should be "ground state"

Figure 5: I am confused about several aspects of the energy level diagrams in panel d. First, what is the unit on the y axis? I assume it's wavenumbers, but there is no axis label. Second, why are there two separate panels? The need for two panels that seem somewhat repetitive is not discussed in the text- if the top one is to be understood as low-Yb content and the bottom is high-Yb content, this should be stated explicitly in the text to distinguish between the two cases. Third, the drawn upwards dotted arrows labelled "energy transfer" in the top panel and "cross-relaxation" in the bottom panel don't make any sense physically. Relaxation and energy transfer should always result in lower energy, not higher energy. Either the physical processes described in these arrows are incorrect or they should be pointing downwards. The discussion of the physical processes in lines 247-256 could also be clearer.

Discussion

Line 264: what is meant by "purification of NPs"?

Line 272-276: Not a complete sentence and it is unclear how this relates to Ref. 36

Figure 6: Caption in (c) should have "luminescence" not "luminescent"

Author Response

Abstract:

Q1. Define RE's (I assume "rare earths") and UC (I assume "up-conversion")

Response: Thank you for your comment. We have changed the term “RE’s” to “rare earths”.

Introduction:

Q2. Line 33: what does "physical bonding" mean? Is this physisorption as opposed to chemisorption?

Response: Thank you for your careful reading. We tried to explain the bonding by electron transport on the surface of nanoparticles.

Q3. Line 51: What is meant by the concerns with fluoride toxicity "as they are less than 980 nm to visible light"? Do the rare-earth fluorides need to be excited by visible light that releases toxic fluoride ions?

Response: Thank you for your careful reading. We tried to describe that the fluorides are excellent host materials in the field of UC phosphors. We have changed the sentences containing the phrase “as they are less than 980 nm to visible light”.

“Although they are the most efficient matrix materials due to their strong UC luminescence intensity and good photostability, some researchers are concerned about the fluoride toxicity of UC phosphors.” : lines 50-51.

Q4. Line 59: with regard to "optical windows", it would be helpful to add why the NIR range is useful in terms of tissue absorption and the potential for in vivo measurements.

Response: Thank you for your careful reading. We have added why the NIR range is useful in terms of tissue absorption and the potential for in vivo measurements.

“Absorption of living cells at near-IR light is low, while that at visible and far-IR light is high owing to hemoglobin and water, respectively. Hemoglobin and water are predominant substances found in large amounts throughout the human body. Therefore, the use of UC nanoparticles can utilize examination of the abdominal cavity through bio-images without requiring a biopsy [6, 19].” : Lines 59-64.

  1. Yagi, Koro, et al. "Preparation of spherical upconversion nanoparticles NaYF4: Yb, Er by laser ablation in liquid and optical properties." Journal of Laser Applications32.2 (2020): 022062.

Q5. Line 67: What are typical surface contaminants and do they cause toxicity or other detrimental effects? It would be good to highlight problems with surface contamination to further motivate the use of PLA for synthesis.

Response: Thank you for your comment. We have added their reasons in this manuscript as this follows.

“Because, during the chemical preparation process, contamination caused by chemical precursors or additives may lead to quenching, surface fouling, and toxicity stem from residuals.” : Lines 72-74.

Q6. Line 69: What does "gravitational atoms" mean?

Response: Thank you for your careful reading. That is typing mistake. We have removed "gravitational".

Q7. Line 74-76: I recommend citing some literature in which PLA nanoparticles are used for biological/medical applications to show that the use of PLA nanoparticles is well established.

Response: Thank you for your comment. We have added some literature in which PLA nanoparticles are used for biological/medical applications.

“It can easily pro-duce nanoparticle-dispersed solutions with high bioaffinity and multielemental compo-site NPs [23, 24].” : Lines 80-81.

  1. Sun, Y.; Zhu, X.; Peng, J.; Li, F. Core–shell lanthanide upconversion nanophosphors as four-modal probes for tumor angiogenesis imaging. Acs Nano 2013, 7, 11290-11300, doi:10.1021/nn405082y.
  2. Liu, Y.; Chen, M.; Cao, T.; Sun, Y.; Li, C.; Liu, Q.; Li, F. A cyanine-modified nanosystem for in vivo upconversion luminescence bioimaging of methylmercury. J. Am. Chem. Soc 2013, 135, 9869-9876, doi:10.1021/ja403798m.

Q8. Line 83: Some examples of "invisible biomolecules" should be provided. As written, it's unclear exactly what UC NPs could detect that other methods can't.

Response: Thank you for your comment. “Invisible biomolecules means that a substance dissolved in a solvent. Thus, we changed the term was modified to “dissolved biomolecules”.:Line 96.

Materials and Methods

Q9. Line 115: "reserve rate" should be changed to "repetition rate"

Response: Thank you for your careful reading. We have changed the “reserve rate” to “repetition rate”.

Q10. Line 116: What is the unit "md" for laser intensity? Usually that should be in W cm^-2 or as fluence in J cm^-2

Response: Thank you for your careful reading. This is a typo. We have changed the "maximum intensity 60 md" to "maximum output of 60 mJ/pulse".

Results

Q11. Line 139: Should add references to the statement about what PLA nanoparticles are used for.

Response: Thank you for your suggestion. We added references related with PLA nanoparticles using in the field of biological application.

  1. Fazio, E.; Gökce, B.; De Giacomo, A.; Meneghetti, M.; Compagnini, G.; Tommasini, M.; & Neri, F. Nanoparticles Engineering by Pulsed Laser Ablation in Liquids: Concepts and Applications. Nanomaterials, 2020, 10(11), 2317, https://doi.org/10.3390/nano10112317

Q12. Line 149: Looking at Figure 2b, it appears that the most up-shifted peak is observed with 1% Yb doping, whereas 3% and 5% are at lower 2theta. Thus the statement that the peak upshifts with increasing Yb content is not consistent with the presented data. Please explain.

Response: Thank you for your careful reading. We are sincerely sorry about this mistake. We have replaced the Figure 2a, b.

Q.13 Figure 3: mean particle size with standard deviation should be added to panels b through e

Response: Thank you for your careful reading. We have replaced the Figure 3 to add the mean particle size with standard deviation.

Q14. Line 173: No NPs >50 nm are included in the size distributions in Figure 3, although some of the TEM images do show them. It would be helpful to clarify that these NPs are rare and motivate why their detailed structure is worth investigating.

Response: Thank you for your suggestion. We have changed Figure 3. (b), (c), (d), (e) to clarify that these NPs are rare. And we have added why it is worth investigating using a literature.

“Minjung Cho et al., showed that a fluorescence dye-labeled various nano sized silica particle suspension was intravenously injected into mice to identify the toxicity, tissue distribution in vivo. Incidence and severity of inflammatory response was transiently increased with injection of 200 and 100 nm silica NPs. But there was no response related to injection of 50 nm particles [25]. Consequently, for biomedical applications, particles smaller than 50 nm are required.” : Lines 179-184.

  1. Cho, M.; Cho, W. S.; Choi, M.; Kim, S. J.; Han, B. S.; Kim, S. H.; Jeong, J. The impact of size on tissue distribution and elimination by single intravenous injection of silica nanoparticles. Toxicol. Lett. 2009, 189, 177-183, doi:10.1016/j.toxlet.2009.04.017.

Q15. Figure 4: What was the Yb doping of the NPs shown in the images?

Response: Thank you for your question. Those TEM images are Er, Yb:Y2O3 NPs with Yb3+ concentration 2 mol%.

Q16. Line 191: Some explanation should be added for why a range of diffraction angles for the (222) plane is observed. Does this have anything to do with the hypothesis that the large NPs come from small NPs being sintered together during PLA?

Response: Thank you for your suggestion and question. It has been reported that the large NPs might have originated from defragmentation from small NPs. Therefore, we have added why a range of diffraction angles for the (222) plane is observed using some literature, and changed this paragraph.

 “The interaction of a laser beam with the surface of the target leads to vaporization of the surface and generates an ablation plume, which contains many species such as atoms, clusters and ions, rushing out with high kinetic energy. After the collapse of the plasma plume, the particles can interact with radiation. This can lead to generation of the large size particles by small nanoparticles as a result of their melting and defragmenting (figure 4c) [31, 32]. As well as, this process can form new compounds containing atoms from both the target and the liquid by these intermediate species in the plume collide [27], and react with molecules of the surrounding liquid through electrons attached and that become negatively charged and attract ions in the plasma [28].” : Lines 216-224.

  1. Svetlichnyi, V. A.; Shabalina, A. V.; Lapin, I. N.; Goncharova, D. A. Metal oxide nanoparticle preparation by pulsed laser ablation of metallic targets in liquid. Applications of Laser Ablation–Thin Film Deposition, Nanomaterial Synthesis and Surface Modification, InTech, Croatia 2016, 245-263, doi:10.5772/65430.
  2. Al‐Mamun, S. A., & Ishigaki, T. Influence of Hydrogen Peroxide Addition on Photoluminescence of Y 2 O 3: Eu 3+ Nanophosphors Prepared by Laser Ablation in Water. J. Am. Ceram. Soc. 2014, 97, 1083-1090, doi:10.1111/jace.12856.

Q17. Line 245: "floor state" should be "ground state"

Response: Thank you for your comment. We have changed the “floor state” to “ground state”.

Q18. Figure 5: I am confused about several aspects of the energy level diagrams in panel d. First, what is the unit on the y axis? I assume it's wavenumbers, but there is no axis label. Second, why are there two separate panels? The need for two panels that seem somewhat repetitive is not discussed in the text- if the top one is to be understood as low-Yb content and the bottom is high-Yb content, this should be stated explicitly in the text to distinguish between the two cases. Third, the drawn upwards dotted arrows labelled "energy transfer" in the top panel and "cross-relaxation" in the bottom panel don't make any sense physically. Relaxation and energy transfer should always result in lower energy, not higher energy. Either the physical processes described in these arrows are incorrect or they should be pointing downwards. The discussion of the physical processes in lines 247-256 could also be clearer.

Response: First, the unit on the y axis IS wavenumbers (cm-1). We are sincerely sorry about this missing.

Second, the first panel shows the mechanism of the possible emissions of Er3+ under 980 nm excitation. When proper Yb3+ is applied, 522 nm and 548 nm emission could be enhanced because of the excited state absorption. However, when the concentration of Yb3+ is increased, the more case of excitation of Yb3+ is occurred, the more the occupying of 4I11/2 in Er3+ is followed, enhancing the 656 nm emission.

Third, the manuscript and figures are edited for the clear description. The point is, excess concentration of Yb3+ results the enhancing 656 nm emission, filling electrons of 4F9/2 level in Er3+, relaxing from 4F7/2 level or exciting from 4F9/2 level.

“Those NPs were analyzed for UC luminescence for visible light areas of Er, Yb:Y2O3 with different Yb3+ concentrations under 980 nm laser in the range of 500 – 700 nm (Figure 5a). Laser irradiation on the specimen produced strong green emissions of 525 and 565 nm and a red emission of 660 nm (Figure 5b). The 525 and 565 nm green emissions correspond to 2H11/2, 4S3/24I15/2 energy transfer of Er3+ ion, and the 660 nm red emission corresponds to 4F9/24I15/2 energy transfer [35] (Figure 5c). This corresponds to an UC process in which Er3+ absorbs 980 nm of low energy and emits 525, 565, and 660 nm of high energy. Doping Yb3+ showed a significant increase in emission intensity in all areas. The green emission was the strongest in the specimen with added 3 mol% Yb3+, but the red emission intensity increased more strongly in proportion to the Yb3+ concentration. This suggests that a high concentration of Yb3+ causes cross relaxation between ions and reduces the green emission intensity [36]. The inset diagram shows the CIE 1931 diagram for the colour change of UC luminescence by Y2O3:Er3+ and Yb3+ NPs excited at 980 nm as the Yb3+ doping concentration increased. As the red element in the luminescence colour became stronger according to the Yb3+ concentration, the luminescence changed from vivid green at 0 mol% to orange at 5 mol%.

The principle of UC emission when 980 nm energy was applied to the Er3+ and Yb3+ ions is shown in Figure 5d. Er3+ ions absorb photons with an energy of 980 nm, resulting in the GSA corresponding to 4I15/24I11/2. Under 980 nm excitation, ESA corresponding to 4I11/24F7/2 by another absorption of 980 nm or multi-phonon relaxation corresponding to 4I11/24I13/2, followed by ESA corresponding to 4I13/24F9/2 could be possible. The electrons excited to 4F7/2 level generate non-radiant relaxation, falling to 2H11/2, 4S3/2, and 4F9/2 level, which causes the emission of energy and produces electromagnetic waves corresponding to 522, 548, and 656 nm, matching with the relaxation to the ground state [37]. In the case of doping Yb3+ ions, the GSA corresponding to 2F7/22F5/2 could be possible under 980 nm excitation. At this point, the energy of Yb3+ can be transferred to Er3+, by relaxing electrons in 2F5/2 to 2F7/2 level. This transferred energy is approximately matched with the excitation of 4I11/24F7/2 or 4I13/24F9/2, enhancing the emission of Er3+ [38]. However, excess concentration of Yb3+ would interrupt the excitation of electrons in Er3+, that is, cross relaxation between 4F7/24I11/2 of Er3+ and 2F7/22F5/2 of Yb3+ could also possible. After then, electrons in 4I11/2 level receives energy from electrons in the 4F7/2 level of other Er3+, resulting in 4I11/24F9/2 and 4F7/24F9/2, respectively. This process finally enhances the 656 nm emission and decrease the intensity of 522 and 548 nm emission [7].” : Lines 238-271.

Discussion

Q19. Line 264: what is meant by "purification of NPs"?

Response: Thank you for your question. Purification of NPs means that only the nanoparticles produced to remove impurities are removed from the solvent.

Q20. Line 272-276: Not a complete sentence and it is unclear how this relates to Ref. 36

Response: Thank you for your question. 

Q21. Figure 6: Caption in (c) should have "luminescence" not "luminescent"

Response: Thank you for your careful reading. We have changed the Figure 6 (c).
